# The role of action potential changes in depolarization-induced failure of excitation contraction coupling in mouse skeletal muscle

**Xueyong Wang[1], Murad Nawaz[1], Chris DuPont[1], Jessica H Myers[1], Steve RA Burke[2], Roger A Bannister[3], Brent D Foy[4], Andrew A Voss[2], Mark M Rich[1]\***

[1]Wright State University, Department of Neuroscience, Cell Biology, and Physiology, Dayton, United States; [2]Wright State University, Department of Biological Sciences, Dayton, United States; [3]University of Maryland School of Medicine, Departments of Pathology/Biochemistry & Molecular Biology, Baltimore, United States; [4]Wright State University, Department of Physics, Dayton, United States

**\*For correspondence:**
mark.rich@wright.edu

**Competing interest:** The authors declare that no competing interests exist.

**Abstract** Excitation-contraction coupling (ECC) is the process by which electrical excitation of muscle is converted into force generation. Depolarization of skeletal muscle resting potential contributes to failure of ECC in diseases such as periodic paralysis, intensive care unit acquired weakness and possibly fatigue of muscle during vigorous exercise. When extracellular $K^+$ is raised to depolarize the resting potential, failure of ECC occurs suddenly, over a narrow range of resting potentials. Simultaneous imaging of $Ca^{2+}$ transients and recording of action potentials (APs) demonstrated failure to generate $Ca^{2+}$ transients when APs peaked at potentials more negative than $-30mV$. An AP property that closely correlated with failure of the $Ca^{2+}$ transient was the integral of AP voltage with respect to time. Simultaneous recording of $Ca^{2+}$ transients and APs with electrodes separated by 1.6mm revealed AP conduction fails when APs peak below $-21mV$. We hypothesize propagation of APs and generation of $Ca^{2+}$ transients are governed by distinct AP properties: AP conduction is governed by AP peak, whereas $Ca^{2+}$ release from the sarcoplasmic reticulum is governed by AP integral. The reason distinct AP properties may govern distinct steps of ECC is the kinetics of the ion channels involved. Na channels, which govern propagation, have rapid kinetics and are insensitive to AP width (and thus AP integral) whereas $Ca^{2+}$ release is governed by gating charge movement of Cav1.1 channels, which have slower kinetics such that $Ca^{2+}$ release is sensitive to AP integral. The quantitative relationships established between resting potential, AP properties, AP conduction and $Ca^{2+}$ transients provide the foundation for future studies of failure of ECC induced by depolarization of the resting potential.

## Editor's evaluation

This is a rigorous electrophysiological paper that investigates relationships between resting potential, action potential properties and conduction and $Ca^{2+}$ transients. It makes an investigation of excitation contraction coupling failure associated with depolarization of the resting potential.

## Introduction

The process by which electrical excitation of muscle is converted into force generation is known as excitation-contraction coupling (ECC). Successful ECC involves propagation of action potentials (APs) from the neuromuscular junction (NMJ) along the length of the fiber as well as into a network of membrane invaginations in muscle known as the transverse tubules (t-tubules) (*Adrian et al., 1969*). Depolarization in the t-tubules activates Cav1.1 channels, which triggers opening of ryanodine receptors, $Ca^{2+}$ exit from the sarcoplasmic reticulum (SR), and force production (*Melzer et al., 1995*; *Dulhunty, 2006*; *Bannister and Beam, 2013*; *Hernández-Ochoa and Schneider, 2018*).

Depolarization of the resting membrane potential of skeletal muscle, when severe enough, causes failure of ECC in diseases such as periodic paralysis and intensive care unit (ICU)-acquired weakness (*Lehmann-Horn et al., 2008*; *Cannon, 2015*; *Friedrich et al., 2015*) as well as potentially contributing to fatigue during intense exercise (*Allen et al., 2008*). Studies of depolarization-induced failure of ECC in frog and mammalian skeletal muscle reveal that whole muscle force is generally stable or slightly increased with mild depolarization of the resting potential, followed by a steep decline with further depolarization by only a few mV (*Holmberg and Waldeck, 1980*; *Renaud and Light, 1992*; *Cairns et al., 1997*; *Yensen et al., 2002*; *Cairns et al., 2011*; *Pedersen et al., 2019*). The decrease in force is paralleled by a decrease in the $Ca^{2+}$ transient with depolarization of the resting potential beyond –60 mV (*Quiñonez et al., 2010*).

The mechanism underlying failure of ECC in the setting of depolarization of the resting potential remains unknown. The simplest possibility is that failure of ECC with depolarization is due to failure to generate or conduct APs. Consistent with this possibility, the decline in force is paralleled by reduction in extracellular recordings of compound muscle APs (*Overgaard et al., 1999*; *Overgaard and Nielsen, 2001*; *Pedersen et al., 2003*). One possible explanation for this correlation is graded failure of excitation with depolarization of the resting potential, which manifests as either a gradual reduction in AP peak or as APs with variable amplitude that increase with increased current injection (*Rich and Pinter, 2001*; *Rich and Pinter, 2003*; *Quiñonez et al., 2010*; *Ammar et al., 2015*; *Miranda et al., 2020*; *Uwera et al., 2020*). Two studies have suggested that reduction in AP peak can cause reduction in force generation (*Cairns et al., 2003*; *Gong et al., 2003*). These studies suggest that partial failure of ECC can occur despite the continued generation of APs.

To determine the mechanism underlying the failure of ECC, we measured generation and conduction of APs as well as the ΔF/F of a genetically encoded $Ca^{2+}$ indicator in muscle fibers in which the resting potential was depolarized by elevation of extracellular $K^+$. The AP parameter that most closely correlated with elevation of intracellular $Ca^{2+}$ was the area of the AP above –30 mV. We further determined that failure of AP propagation occurs when APs peak below –21 mV. Identification of the AP properties governing conduction and $Ca^{2+}$ release along the length of the fiber provides a basis for future studies of depolarization-induced failure of ECC.

## Results

### Failure of force generation correlates with failure of the $Ca^{2+}$ transient

To confirm the $K^+$ concentration dependence of force generation previously reported (*Cairns et al., 1997*; *Yensen et al., 2002*; *Pedersen et al., 2019*), we perfused solution containing elevated concentrations of $K^+$ and measured force. Measurement of twitch force in the mouse extensor digitorum longus (EDL) following elevation of extracellular $K^+$ revealed an initial increase in force (*Figure 1A*, n = 3 muscles for each $K^+$ concentration), which was followed by a decline that became faster with higher levels of extracellular $K^+$ (*Figure 1A*). With return to solution containing normal $K^+$, force recovered rapidly. The $K^+$ concentration dependence of force 40 min after $K^+$ infusion was steep: between 10 and 14 mM, force went from near normal to almost 0 (*Figure 1A*). These findings agree well with previous studies (*Cairns et al., 1997*; *Yensen et al., 2002*; *Pedersen et al., 2019*).

To determine the dependence of force on resting potential, a separate set of experiments were done on a different set of muscles in which intracellular recording was performed 20–60 min following infusion of high $K^+$. The difference in resting potential between the various concentrations of $K^+$ was modest: in 10 mM $K^+$, where force was near normal, Rm averaged –62.3 ± 1.0 mV (n = 5 mice, 50 fibers) and in 14 mM $K^+$, where force had largely failed it averaged –57.1 ± 1.5 mV (n = 4 mice, 32 fibers). These data suggest that a steep reduction of force is caused by depolarization over a narrow

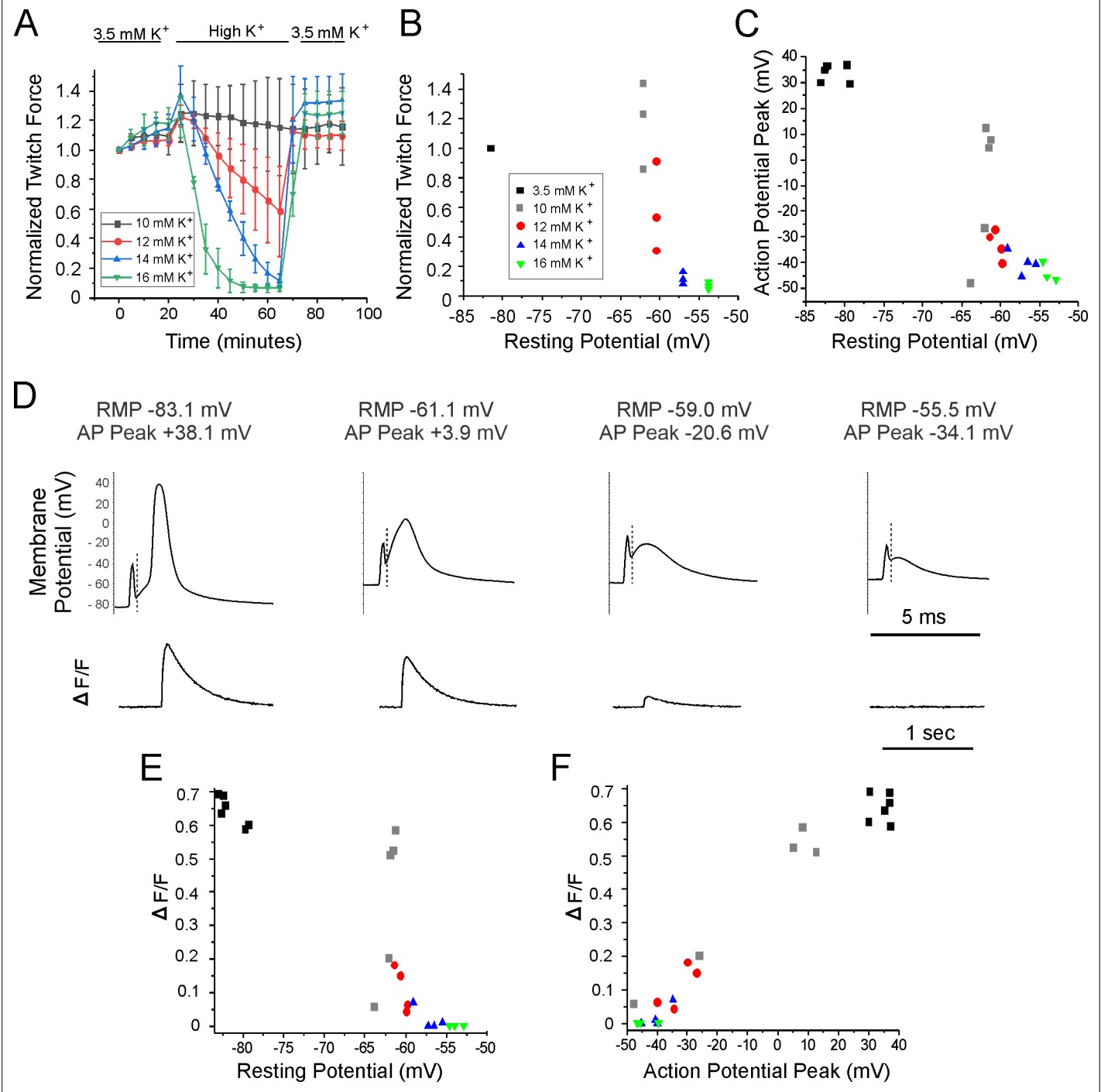

**Figure 1.** Relationships between resting potential, action potential (AP) peak, ΔF/F, and force generation. (**A**) A plot of extensor digitorum longus (EDL) twitch force versus time following infusion of various concentrations of external K[+] (n = 3 muscles for each K[+] concentration, error bars = SD). Force for each muscle was normalized to the initial force. (**B**) Individual muscle forces after 40 min in high K[+] (same experiments used to generate data in **A**) normalized to baseline force in 3.5 mM K[+] for the same muscle, plotted versus mean resting membrane potential recorded in a separate set of experiments 20–40 min following infusion of high K[+] solution (for resting potential measurements n is the following: 3.5 mM K[+]: 6 muscles, 49 fibers, 10 mM K[+]: 5 muscles, 50 fibers, 12 mM K[+]: 4 muscles, 36 fibers, 14 mM K[+]: 4 muscles, 32 fibers, 16 mM K[+]: 3 muscles, 32 fibers). The external K[+] concentration is indicated by the symbol used for each muscle. (**C**) Plot of mean AP peak for individual muscles versus mean resting potential for the same muscle at K[+] concentrations ranging from 3.5 mM to 16 mM. The plot was generated using the same muscles studied to generate the mean resting potential plotted in (**B**). (**D**) The APs and the corresponding change in fluorescence signal (ΔF/F) for fibers with various resting potentials. A dotted vertical line has been drawn on each voltage trace to mark the end of the stimulus artifact. Note the difference in time scales for the AP and ΔF/F

*Figure 1 continued on next page*

*Figure 1 continued*

traces. (**E**) Plot of mean ΔF/F versus mean resting potential for each muscle studied. (**F**) Plot of mean ΔF/F versus mean AP peak for each of the muscle studied.

range of resting potentials (*Figure 1B*), similar to what has been reported previously (*Cairns et al., 1997*).

To determine the mechanism underlying depolarization-induced failure of force generation, APs were recorded. AP amplitude decreased as the resting potential became more depolarized with elevation of extracellular K+ (*Figure 1C and D*). What was somewhat surprising was that small APs could still be triggered in many fibers at K+ concentrations (14 and 16 mM) at which force production was near 0. These data suggest that failure of force production is not due to failure to initiate APs.

To examine whether failure of force could be explained by failure of Ca$^{2+}$ transients, we simultaneously recorded APs while imaging Ca$^{2+}$ in mice expressing GCAMP6f in skeletal muscle. These experiments were performed on the same muscles used to measure resting potential and AP properties following infusion of solutions containing elevated concentrations of K+. GCAMP6f is a high-affinity Ca$^{2+}$ indicator with a Kd near 600 nM in cardiomyocytes such that it can detect changes in Ca$^{2+}$ ranging from 100 nM to 5 µM (*Shang et al., 2014*). The relatively high affinity makes it possible to measure changes in signal (ΔF/F) during low release flux events. However, due to the high affinity of GCAMP6f it is not possible to study kinetics of the underlying Ca$^{2+}$ signal (*Chen et al., 2013*; *Shang et al., 2014*).

APs were triggered by 0.2 ms injections of current in various concentrations of K+ (*Figure 1D*), and the ΔF/F triggered by the AP was recorded (*Figure 1D*). ΔF/F in each fiber was normalized to the maximal ΔF/F for that fiber to allow determination of how effective the AP was at triggering Ca$^{2+}$ release. The maximal ΔF/F for each fiber was obtained using a 20 ms injection of current, which depolarized the fiber to 0 mV or more locally. 0 mV was chosen as depolarization during APs to above 0 mV triggers minimal further increases in ΔF/F (*Figure 1F*).

In normal extracellular K+, single APs generated a robust signal ΔF/F (*Figure 3—video 1* associated with this submission). The normalized ΔF/F in 10 mM K+ was highly variable, both between fibers and muscles. In 3/5 muscles, ΔF/F was similar to ΔF/F at a resting potential of –85 mV in 3.5 mM K, but in two muscles it was lower. The plot of ΔF/F versus resting potential was similar to the plot obtained for muscle force versus resting potential (*Figure 1E*). These data suggest that sudden failure to trigger Ca$^{2+}$ release from the SR is the mechanism underlying the sudden depolarization-induced failure of ECC.

We next examined whether failure to generate APs was the mechanism underlying failure to trigger Ca$^{2+}$ release from the SR. When extracellular K+ was 3.5 mM, AP peak averaged +30 to + 40 mV. With elevation of K+ to 10, 12, 14, and 16 mM, AP peaks ranged from near +10 mV to –40 mV (*Figure 1C and D*). When APs peak between –20 and –30 mV, small ΔF/F signals could be detected and when APs peak below –30 mV, ΔF/F was 0 (*Figure 1D and F*). These data suggest that failure of the Ca$^{2+}$ transient and force generation occurs prior to failure to initiate APs.

## Failure of ECC in individual fibers

To obtain a more detailed understanding of the relationships between resting potential, AP peak, and ΔF/F, we recorded from individual fibers for 7 min during infusion of solution containing 16 mM K+. When extracellular K+ was kept at 3.5 mM, the ΔF/F was stable over time, with a slight trend towards increasing (*Figure 2A and B*, 1.16 at 7 min vs. initial value normalized to 1, p=0.07, n = 5 fibers).

With infusion of 16 mM K+, depolarization of the resting membrane potential was accompanied by graded reduction in the AP peak from a maximum ranging from +15 to + 35 mV down to –25 to –45 mV (*Figure 2C, D and E*). Between resting potentials of –65 to –52 mV, there was rapid reduction of the peak with further depolarization. To quantitate the relationship between depolarization of the resting potential and reduction in AP peak, we fit the data for individual fibers with Boltzmann equations. For these fits, the HV limit, which represents the minimal AP peak when resting potential was elevated, was constrained to be between –30 mV and –50 mV. The V50 for the resting potential at which AP peak was half maximal was –58.2 ± 3.3 mV, the slope factor *k* was 1.8 ± 0.6 mV, and the average value of the half-maximal AP peak at the V50 value was –15.5 ± 4.9 mV (n = 12 fibers from six mice).

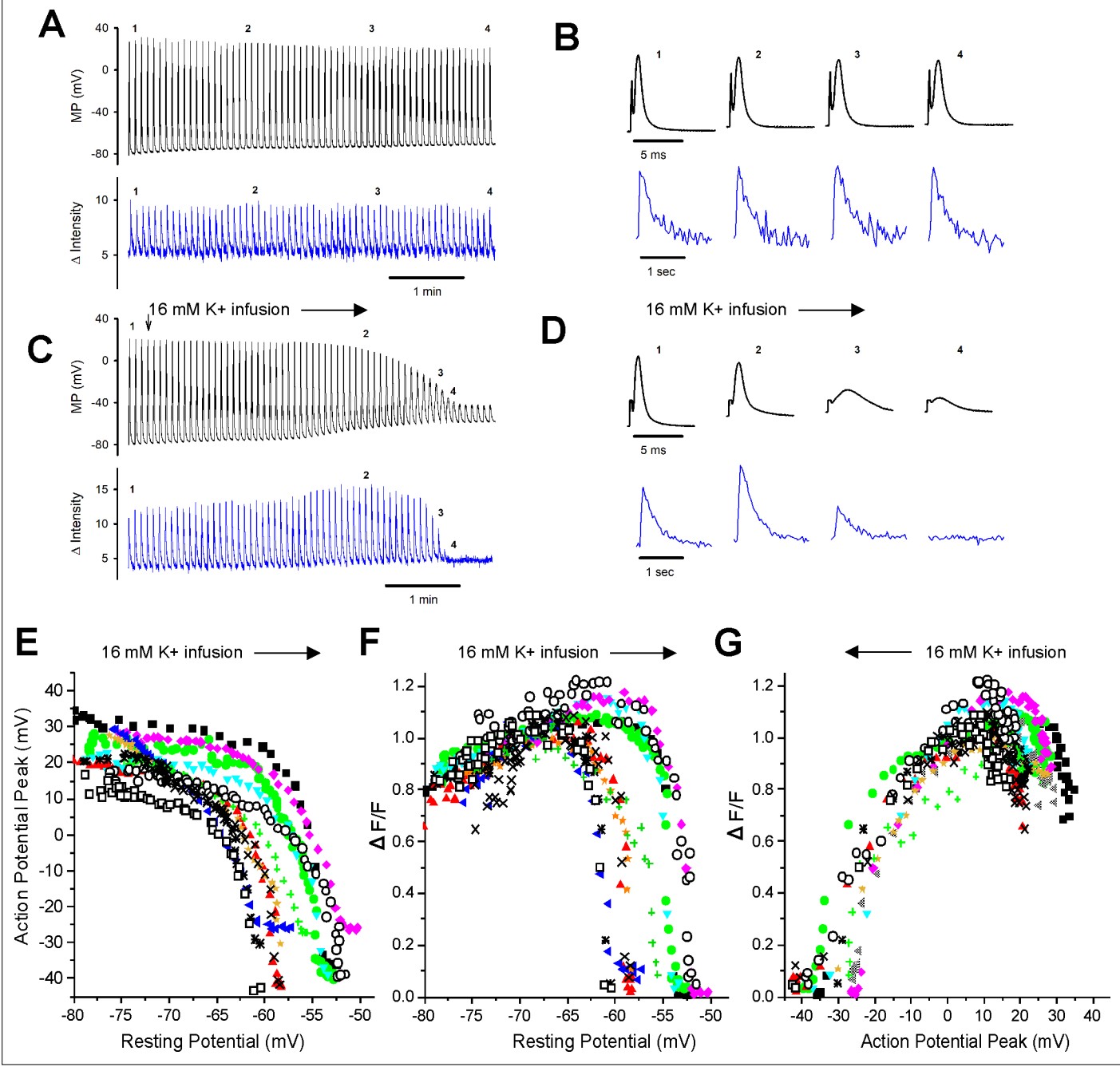

**Figure 2.** Failure of the intracellular Ca$^{2+}$ transient in individual fibers during depolarization of the resting potential. (**A**) The action potentials (APs) and ΔF/Fs during a 7 min recording for a fiber expressing GCAMP6f in 3.5 mM K$^+$. The stimulus artifact in the AP traces has been eliminated for clarity. The trace shown is not continuous. A 5 ms block of time is shown for each AP, and a 1 s block of time is shown for each ΔF/F. The time base indicated is for the time between APs and ΔF/F. (**B**) The APs and corresponding ΔF/F for the four time points indicated in (**A**) on an expanded time scale. (**C**) APs and ΔF/F for a fiber during infusion of solution containing 16 mM K$^+$ (indicated by the vertical and horizontal arrows). (**D**) The APs and corresponding ΔF/F for the four time points indicated in (**C**) on an expanded time scale. (**E**) Plot of AP peak versus resting potential during infusion of 16 mM K$^+$ for 12 fibers. (**F**) Plot of ΔF/F versus resting potential for the 12 fibers shown in (**E**). The ΔF/F present at a resting potential of –70 mV was normalized to a value of 1 for each fiber. (**G**) Plot of normalized ΔF/F versus AP peak for the 12 fibers studied.

With depolarization during infusion of 16 mM K$^+$, there was an initial increase in the ΔF/F from a mean normalized value of 0.81 ± 0.10 to the normalized maximum of 1 (p<0.0001 paired $t$-test, n = 12, *Figure 2D and F*). This increase is similar to what has been reported previously (*Quiñonez et al., 2010*; *Pedersen et al., 2019*) and occurred despite reduction in AP peak (*Figure 2C and E*).

As depolarization progressed, there was rapid, complete loss of ΔF/F (*Figure 2C, D and F*). To quantitate the relationship between depolarization and failure of the Ca²⁺ transient, we fit the data for the decrease in ΔF/F with depolarization of the resting potential beyond –70 mV with a Boltzmann equation (*Equation 1*). For these fits, the LV limit, which represents the ΔF/F when resting potential was –70 mV, was fixed to 1, and the HV limit was constrained to be between 0 and 0.1. The resting potential at which ΔF/F was half maximal was –57.5 ± 3.4 mV with a slope factor of 0.4 ± 0.2 mV (*Figure 2F*), which was significantly steeper than the slope for reduction in AP peak (p<1 × 10⁻⁵, paired *t*-test, n = 12 fibers from five mice).

We plotted the reduction in ΔF/F versus AP peak (*Figure 2G*) and fit the data with a Boltzmann equation (*Equation 1*). For these fits, the HV was fixed to 1 and the LV was fixed to 0. The ΔF/F was half maximal at an AP peak of –21.0 ± 4.0 mV with a slope factor of 5.9 ± 1.7 mV. This relationship between peak voltage and Ca²⁺ transient was within the range of values obtained from voltage-clamp studies of mouse muscle fibers (*Wang et al., 1999*; *Ferreira Gregorio et al., 2017*). These data suggest that APs peaking below –30 mV trigger little to no elevation of intracellular Ca²⁺ and hence generate little to no force.

## AP integral correlates with the Ca²⁺ transient

To identify properties of APs that would allow for accurate prediction of ΔF/F, the correlation between normalized AP amplitude and ΔF/F was examined. A loss of 11.6 ± 1.7 mV of resting potential was required to reduce AP amplitude from 90% to 10% of maximum. In contrast, ΔF/F was reduced from 90% to 10% of maximum with a loss of only 4.1 ± 2.4 mV of resting potential (p<1 × 10⁻⁶ vs. APs, n = 12, paired Student's *t*-test). This difference led us to look for another AP parameter that more closely correlated with the reduction in ΔF/F.

As shown in *Figure 2G*, an AP peak above –30 mV is required to consistently trigger ΔF/F. We thus set AP peaks of –30 mV or below to 0 and normalized AP amplitude. At mildly depolarized resting potentials, drops in AP peaks were accompanied by increases in the ΔF/F (*Figure 2D and F*; *Quiñonez et al., 2010*; *Pedersen et al., 2019*). At more depolarized resting potentials, the drop in AP peak correlated with the drop in ΔF/F (*Figure 2G*); the mean R² was 0.65 ± 0.14. While this indicated a reasonable linear correlation between AP peak and ΔF/F, we wished to identify a parameter that more closely correlated with ΔF/F.

To include changes in both AP half width and peak, the integral of AP voltage with respect to time was measured. The integral above –30 mV was used to account for the lack of ΔF/F when APs peaked below –30 mV. AP integral closely paralleled ΔF/F during depolarization of the resting potential (*Figure 3A and B*). When the normalized ΔF/F was plotted against normalized AP area, the mean R² value was 0.86 ± 0.11 (*Figure 3C*, p<0.001 vs. the R² value for ΔF/F vs. AP peak, n = 12 fibers, paired *t*-test). The R² value was larger for AP area vs. ΔF/F because AP area more closely mimicked the rapid decrease in ΔF/F (*Figure 3B*).

## Failure of AP propagation in individual fibers

The correlation between Ca²⁺ transient and integrated AP peak above a threshold suggests a mechanism by which force generation can fail even in the presence of APs. Another possibility is that small APs triggered by depolarization may fail to propagate. To measure AP propagation, fibers were imaged with a ×5 objective and the current injecting and membrane potential measuring electrodes were separated by 1.6 mm. Impalement of the same fiber was achieved using the fluorescent signal generated by fibers expressing GCAMP6f (*Figure 4A*). The length constant of mouse EDL fibers is close to 0.5 mm (*Riisager et al., 2014*) such that depolarization due to passive spread of current from the current injecting electrode would be expected to be a minor contributor to depolarization of the membrane potential at the recording electrode. We confirmed that the 0.2 ms current injection used to trigger APs did not spread to the recording electrode: injection of 1000 nA of hyperpolarizing current for 0.2 ms caused no detectable voltage deflection (data not shown).

With electrodes separated by 1.6 mm, we observed two patterns of AP failure. In some fibers, there was all or none failure, with sudden drops in AP peak of 30–60 mV (*Figure 4B and E*). In others, reduction of AP peak was continuous (*Figure 4B and C*). Regardless of whether reduction in AP peak was sudden or continuous, it was steeply dependent on resting potential; with a >40 mV reduction in peak occurring over 1–2 mV of resting potential (*Figure 4B*). This is in contrast to the more gradual

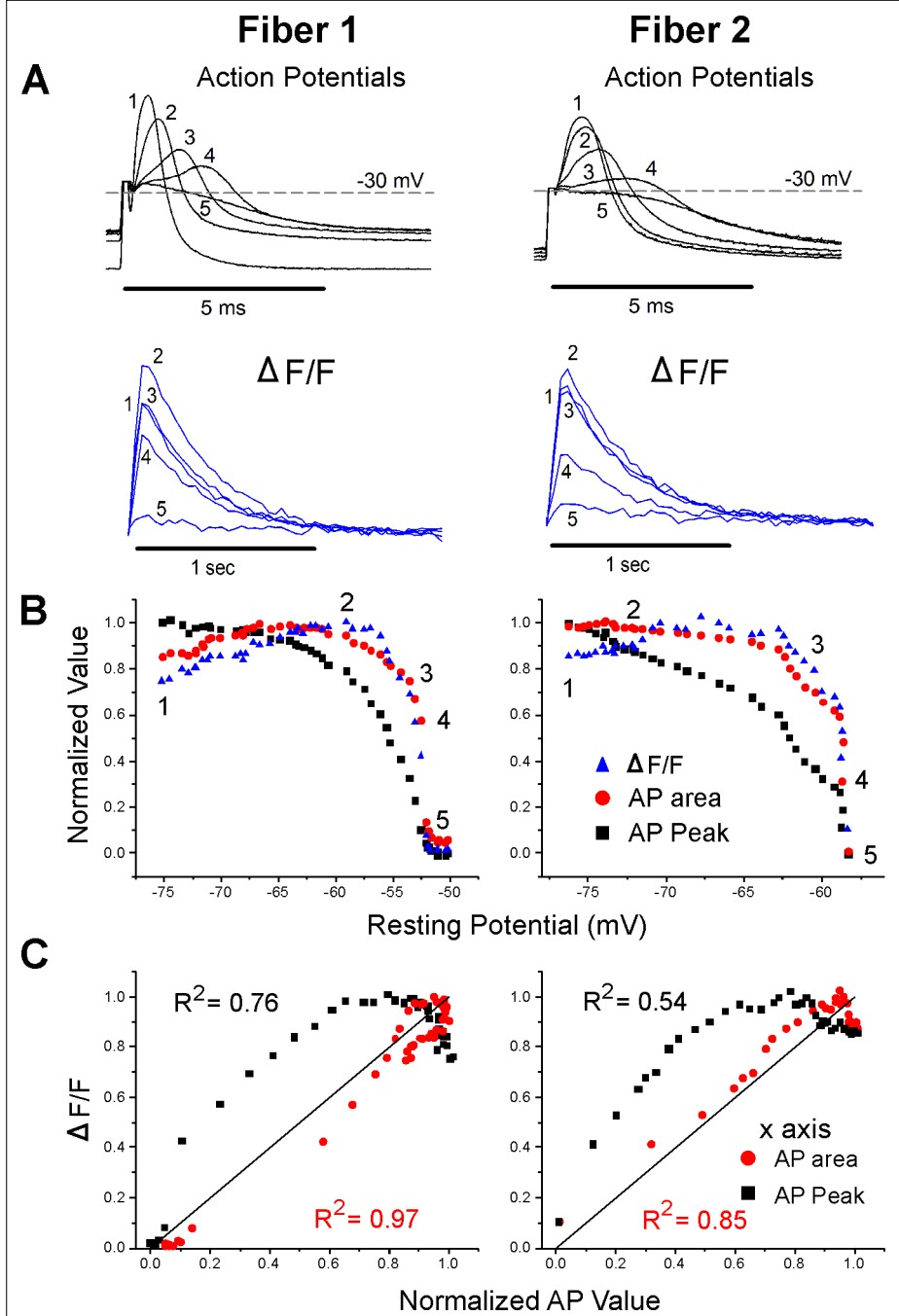

**Figure 3.** Correlation between ΔF/F and action potential (AP) area. (**A**) The AP traces and ΔF/F recorded from two muscle fibers during infusion of 16 mM K+. A horizontal line at –30 mV represents the cutoff for measurement of AP area and normalized AP peak. APs peaking below –30 mV had peaks and areas set to 0. Below the AP, traces are the corresponding ΔF/F. The stimulus artifact has been truncated in the AP traces for clarity. (**B**) Plots of normalized AP peak, normalized AP area, and normalized ΔF/F versus resting potential for two fibers. The numbers 1–5 on each plot represent the points corresponding to the five AP and ΔF/F traces shown in (**A**). (**C**) Plots of the normalized ΔF/F versus either AP area or AP peak for the two fibers in (**A**) and (**B**). The line of identity is drawn on each plot as a reference. The R² value for each relationship is shown on the graph.

The online version of this article includes the following video for figure 3:

**Figure 3—video 1.** Failure of the Ca²⁺ transient in a mouse extensor digitorum longus (EDL) fiber during infusion of 16 mM K+.

https://elifesciences.org/articles/71588/figures#fig3video1

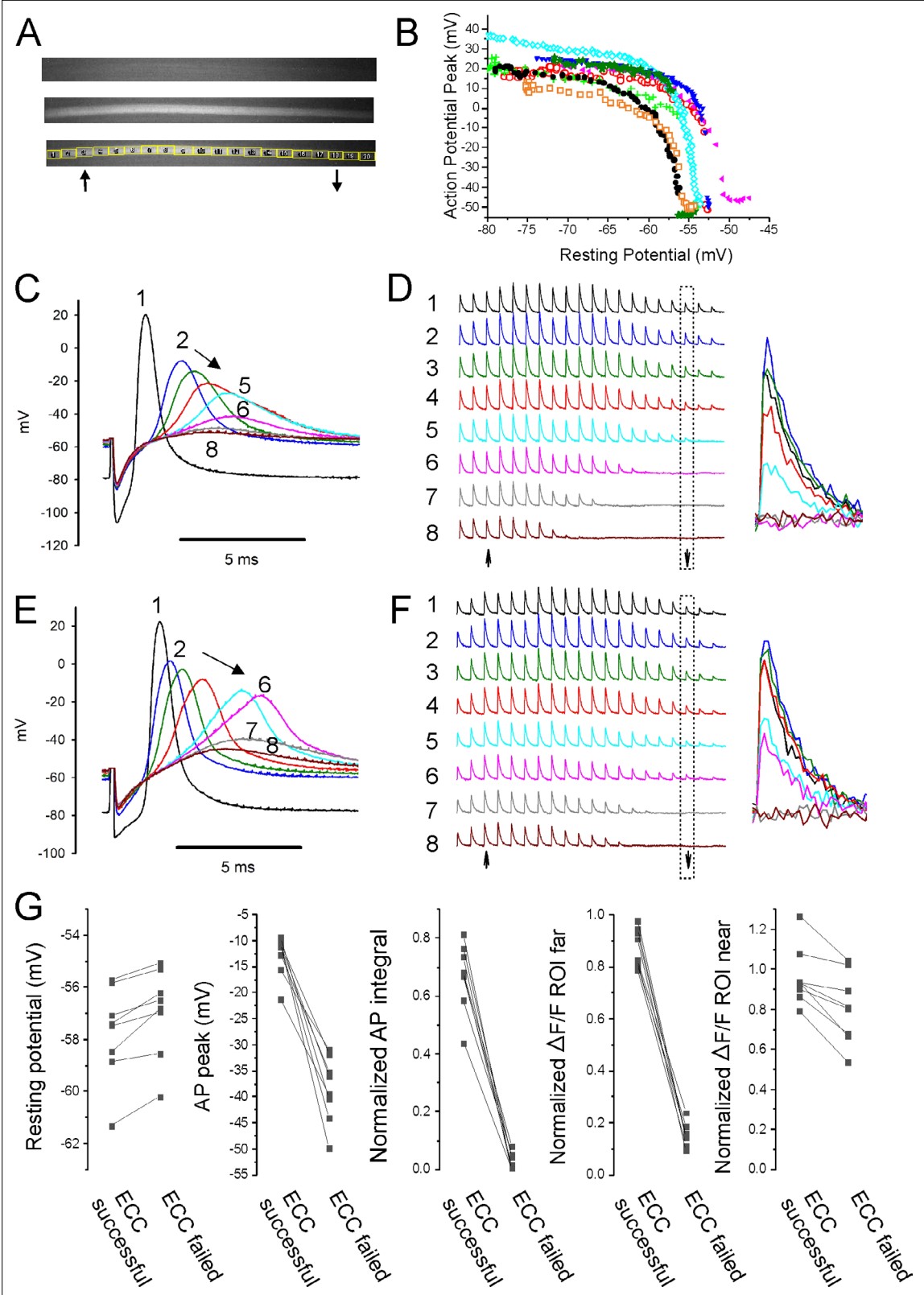

**Figure 4.** Failure of action potential (AP) conduction contributes to depolarization-induced failure of excitation-contraction coupling (ECC). (**A**) A fiber expressing GCAMP6f at 5×. The top image is the fiber at rest. The middle image shows the signal along the length of the fiber triggered by an AP. The bottom image shows the regions of interest (ROIs) placed along the length of the fiber. The up arrow indicates the position of the stimulating electrode, and the down arrow indicates the position of the recording electrode. (**B**) Plot of AP peak 1.6 mm from the stimulating electrode versus resting potential

*Figure 4 continued on next page*

*Figure 4 continued*

during infusion of 16 mM $K^+$ for eight representative fibers from eight different muscles. (**C**) Eight superimposed APs from a fiber during infusion of 16 mM $K^+$. (**D**) The $\Delta F/F$ at the 20 ROIs along the length of the fiber in (**C**) for each of the eight APs shown in (**C**). The upward-pointing arrow indicates the position of the stimulating electrode, the downward-pointing arrow indicates the position of the recording electrode. The dashed boxes indicate the $\Delta F/F$ at the ROI of the recording electrode. At the right are shown the eight superimposed $\Delta F/Fs$ for the ROI at the recording electrode. In the example shown, in stimulations 1–3 ECC was judged to have succeeded, in stimulations 4 and 5 it was indeterminate, and in stimulations 6–8 it failed. (**E, F**) APs and the corresponding $\Delta F/F$ for a second fiber. In (**F**), it was judged that in stimulations 1–4 ECC was successful, in 5 and 6 it was indeterminate, and 7 and 8 it failed. (**G**) Plots of muscle means from eight mice for parameters when ECC succeeded versus when ECC failed. Each point represents an average of 2–6 muscle fibers from a muscle. For plots of $\Delta F/F$, ROI far was located at the recording electrode and ROI near was located at the stimulating electrode.

reduction in AP peak seen with depolarization when the electrodes were close together (*Figure 2C and E*).

To determine the characteristics of APs necessary for successful ECC along the length of fibers, regions of interest (ROIs) were used to track $Ca^{2+}$ release at various distances from the stimulating electrode during infusion of 16 mM $K^+$. At ×5 magnification, intensity of illumination was not uniform across the muscle (*Figure 4A*). Despite the variation in illumination, the signal-to-noise ratio at each ROI allowed for analysis of $\Delta F/F$ along the length of fibers. APs at the recording electrode (1.6 mm from the stimulating electrode) and $\Delta F/F$ at 20 ROIs along the length of individual fibers were followed (*Figure 4C–F*). The $\Delta F/F$ at each ROI was normalized to the value when the resting potential was –70 mV to account for the gradient in brightness. ECC was considered to have been successful if $\Delta F/F$ of the ROI at the recording electrode was ≥75% of the $\Delta F/F$ for that ROI at a resting potential of –70 mV. ECC was considered to have failed when $\Delta F/F$ of the ROI at the recording electrode fell to ≤25% of the $\Delta F/F$ at –70 mV. When $\Delta F/F$ was between 75 and 25% of the value at a resting potential of –70 mV, success of ECC was considered indeterminate, and the data were not analyzed (*Figure 4C–F*).

The last AP to trigger successful ECC was associated with a $\Delta F/F$ at the ROI 1.6 mm from the stimulating electrode of 87% ± 7% of the $\Delta F/F$ value for the ROI at a resting potential of –70 mV, and this fell to 16% ± 4% by the first AP that failed to trigger successful ECC (*Figure 4G*, n = 8 muscles). The difference in resting potential between the last successful ECC and the first failed ECC was 0.8 ± 0.5 mV (*Figure 4G*, n = 8 muscles). The AP peak of the last successful ECC averaged –12.8 ± 3.9 mV and by the first failed ECC it had dropped to –38.5 ± 6.3 mV (*Figure 4G*, n = 8 muscles). The normalized integral of the last AP to successfully trigger ECC averaged 0.67 ± 0.12, and this fell to 0.03 ± 0.03 for the first failed ECC. The normalized integrals of APs that triggered successful ECC along the length of the fiber ranged from 0.43 to 0.81 (*Figure 4G*). The peaks of the last APs to trigger successful ECC along the length of the fiber ranged from –9.2 to –21.2 mV (*Figure 4G*). Neither measure has any overlap in values between successful and failed ECC. These data suggest that AP peak is as good as AP integral at predicting failure of conduction.

At the time of failure of AP conduction, $\Delta F/F$ near the stimulating electrode still averaged 96% ± 15% of the $\Delta F/F$ at that ROI at a resting potential of –70 mV (*Figure 4G*). This resulted in a gradient of $\Delta F/F$ along the length of the fiber (*Figure 4D and F*). These data suggest that local APs, which fail to conduct along the length of fibers, can be large enough to locally invade t-tubules, trigger gating of Cav1.1 channels and $Ca^{2+}$ release from the SR.

To confirm that APs generated at the current injecting electrode sometimes failed to conduct, fibers were imaged at 5× and the stimulating and recording electrodes were placed within 100 μm of each other. APs and $\Delta F/F$ along the lengths of fibers were followed during infusion of 16 mM $K^+$ (*Figure 5A–D*). It was assumed that APs had successfully conducted if $\Delta F/F$ at the ROI at the edge of the image was >0.75 of its value at a resting potential of –70 mV. Similar to recordings with the electrodes far apart, the average depolarization of resting potential between successful and failed AP conduction was only 0.5 ± 0.2 mV (n = 10 muscles with 2–5 fibers, *Figure 5E*). The peaks of APs that conducted ranged from –11.9 to –20.4 mV (*Figure 5E*). This range is similar to the range of AP peaks recorded with electrodes far apart. Normalized AP integral of successfully propagating APs ranged from 0.38 to 0.83 (*Figure 5E*). We conclude that, just like with electrodes far apart, AP peak is as good a predictor of AP conduction as AP integral. If APs peak above –21 mV, they conduct along the length

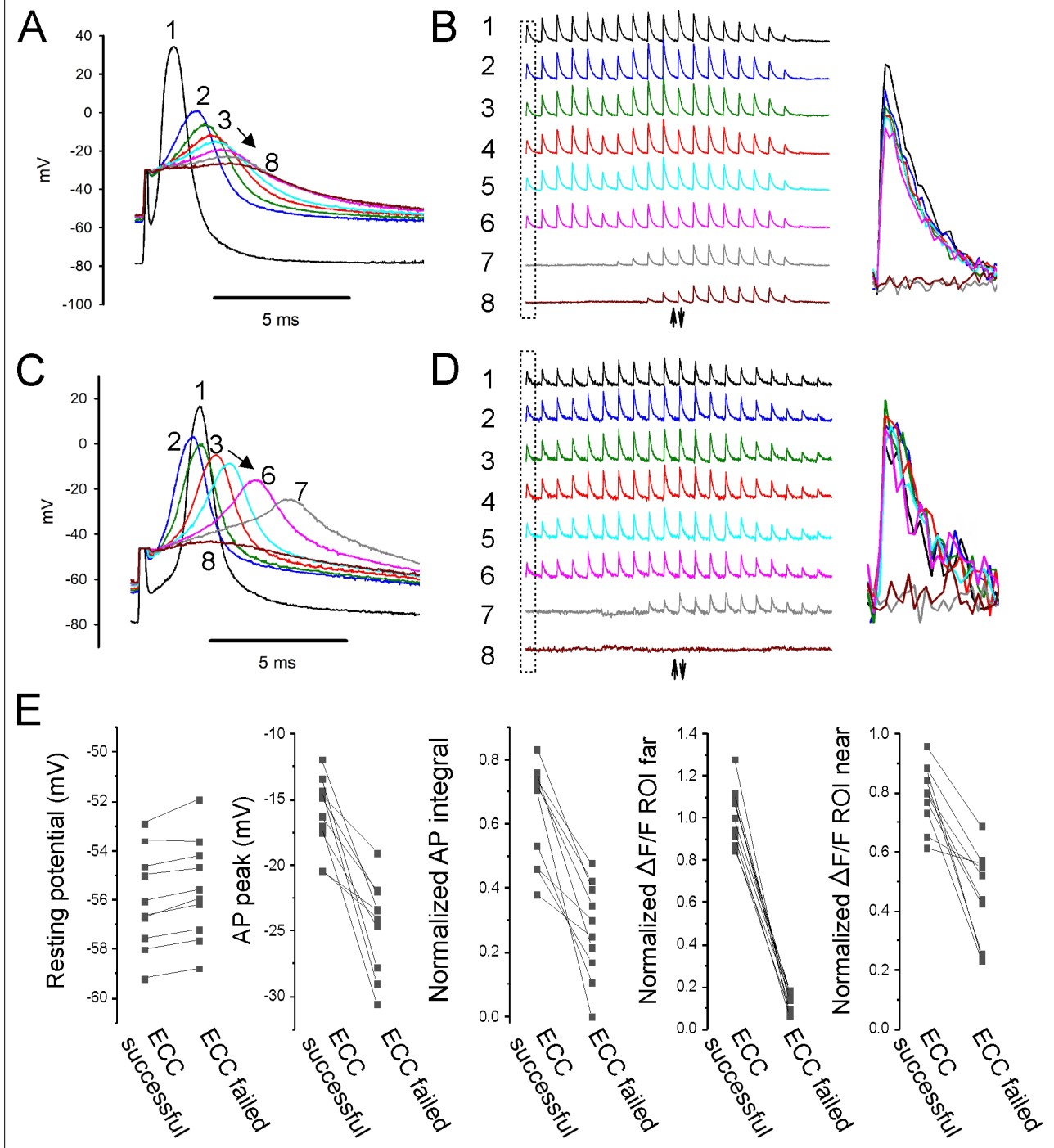

**Figure 5.** Properties of action potentials (APs) near the stimulating electrode that predict successful excitation-contraction coupling (ECC). (**A, C**) Eight superimposed, selected APs from two fibers during infusion of 16 mM K[+]. (**B, D**) The ΔF/F at the 20 regions of interest (ROIs) along the length of the fibers for each of the eight APs shown in (**A, C**). The upward-pointing arrow indicates the position of the stimulating electrode, the downward-pointing arrow indicates the position of the recording electrode. The dashed boxes indicate the ΔF/F furthest from the stimulating electrode. At the right are shown the eight superimposed ΔF/Fs for the ROI furthest from the stimulating electrode. In both (**B**) and (**D**), ECC was judged to have succeeded in stimulations 1–6, there were no indeterminate stimulations, and ECC was judged to have failed in stimulations 7 and 8. (**E**) Plots of muscle means from 10 mice for parameters when EEC succeeded versus when ECC failed. Each point represents an average of 2–5 muscle fibers from a muscle.

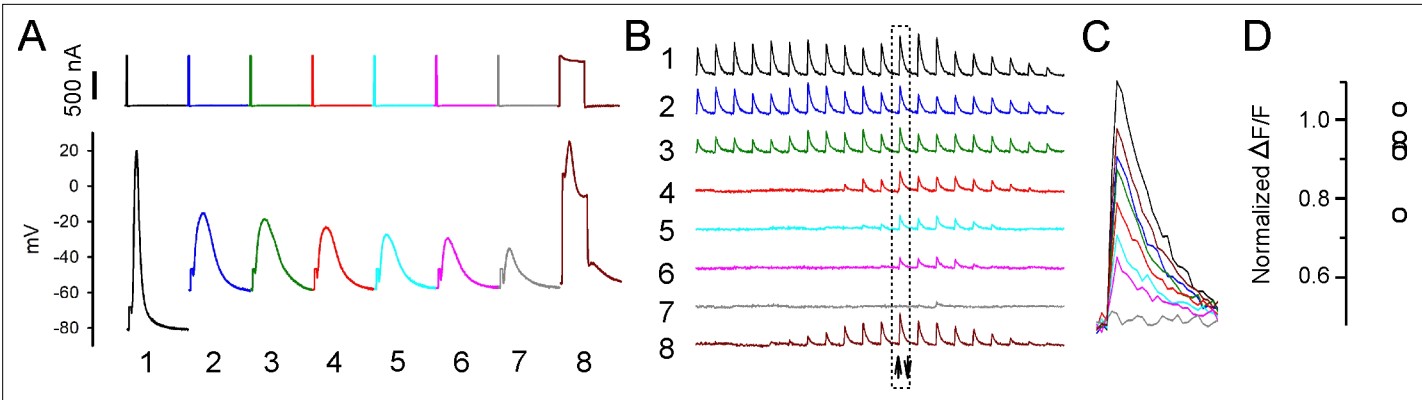

**Figure 6.** Sarcoplasmic reticulum (SR) $Ca^{2+}$ release can be evoked after depolarization-induced failure of excitation-contraction coupling (ECC).
(**A**) Shown are eight current injections and the corresponding action potentials (APs) in a fiber during infusion of 16 mM $K^+$. Following >90% reduction in ΔF/F at the site of current injection, the duration of the current injection was increased to 5 ms to trigger an artificial 'AP' in stimulation #8. The 'AP' peak triggered is similar to the peak of the AP in that fiber prior to infusion of 16 mM $K^+$ (stimulation #1). (**B**) The ΔF/F at the 20 regions of interest (ROIs) along the length of the fiber for each of the eight APs shown in (**A**). The upward-pointing arrow indicates the position of the stimulating electrode, the downward-pointing arrow indicates the position of the recording electrode. The dashed boxes indicate the ΔF/F closest to the stimulating electrode. (**C**) Eight superimposed ΔF/Fs for the ROI closest to the stimulating electrode. In the example shown, the 5 ms injection of current triggered a ΔF/F that was 87% of that triggered by an AP in that fiber at a resting potential of –70 mV. (**D**) Plot of muscle means from five muscles for ΔF/F triggered by a 5 ms current injection normalized to ΔF/F for APs from the same fibers at a resting potential of –70 mV.

of the fiber without decrement. APs peaking below –21 mV can trigger local $Ca^{2+}$ release, but fail to conduct.

## The role of Nav1.4 and Cav1.1 inactivation in depolarization-induced failure of ECC

When severe enough, depolarization of the resting membrane potential triggers both inactivation of Nav1.4 channels and inactivation of $Ca^{2+}$ release from the SR (*Ferreira Gregorio et al., 2017*; *Cannon, 2018*; *Hernández-Ochoa and Schneider, 2018*). The trigger for $Ca^{2+}$ release from the SR is gating charge movement of Cav1.1 channels, which has a midpoint of inactivation of –57 mV (*Ferreira Gregorio et al., 2017*); a membrane potential at which ΔF/F sharply decreases.

To determine whether inactivation of Nav1.4 channels and/or inactivation of SR $Ca^{2+}$ release is the primary mechanism underlying depolarization-induced failure of ECC, we examined whether SR $Ca^{2+}$ release could still be triggered after ECC had failed. For these experiments, recordings were continued during infusion of 16 mM $K^+$ until AP amplitude in the region of the current injecting electrode was below –30 mV such that there was a >90% reduction of the localized ΔF/F near the current injecting electrode (*Figure 6A and B*). Following failure of ECC, the duration of current injection was increased to 5 ms to provide an artificial 'action potential.' Current injected was increased until it was sufficient to depolarize fibers to between +10 and +35 mV (*Figure 6A*). This range of voltages was selected as it is the range of AP peaks observed when the resting potential is near –80 mV (*Figure 2E*).

When the duration of current injection was increased to 5 ms, a 10–20 mV depolarization caused by opening of Na channels was superimposed on the depolarization caused current injection (*Figure 6A*). The ΔF/F triggered by the artificial 'AP' was normalized to the ΔF/F in that fiber triggered by an AP when resting potential was –70 mV. The mean 'AP' peak triggered by 5 ms current injection was 26.2 ± 6.3 mV (n = 13 fibers from five muscles) and was triggered at a mean resting potential of –51.6 ± 4.6 mV. The ΔF/F triggered by the 'AP' averaged 92% ± 10% of the ΔF/F triggered by APs from the same fiber at a resting potential of –70 mV (*Figure 6C and D*). While these data do not rule out partial inactivation of SR $Ca^{2+}$ release, they suggest that following depolarization of the resting potential to –52 mV there is, at most, modest inactivation of SR $Ca^{2+}$ release. We conclude that inactivation of Nav1.4, and the resultant failure of APs to sufficiently depolarize the fiber, is the primary mechanism underlying depolarization-induced failure of ECC.

## Discussion

While the steps underlying ECC in skeletal muscle have been well studied, the properties of APs necessary for successful ECC have not been determined. Elevation of extracellular $K^+$ causes AP peaks to vary by 70 mV in amplitude, ranging from +30 to –40 mV. We combined intracellular recordings of APs at different distances from the stimulating electrode with $Ca^{2+}$ imaging to determine the AP characteristics necessary for successful conduction and ECC. Ours is the first study to determine quantitative relationships between resting potential, APs, AP propagation, and the $Ca^{2+}$ transients that trigger contraction during ECC in skeletal muscle. Understanding these relationships provides the foundation for future studies of depolarization-induced failure of ECC in various disease states such as periodic paralysis, ICU-acquired weakness, and perhaps fatigue during strenuous exercise.

### APs peaking above –21 mV propagate

Several studies have suggested that failure of AP conduction can be an important contributor to failure of ECC in certain situations. Conduction velocity of mouse EDL muscle fibers has previously been measured using a three-electrode intracellular recording set up and by using a $Ca^{2+}$ indicator dye (*Riisager et al., 2014*; *Banks et al., 2018*). Declines in AP peak and AP area and the resultant failure to conduct along the length of muscle fibers were found during prolonged repetitive stimulation of fibers (*Riisager et al., 2014*). Nonpropagating $Ca^{2+}$ release triggered by depolarization at the end of fibers during bipolar extracellular stimulation has also previously been demonstrated (*Hernández-Ochoa et al., 2016*; *Banks et al., 2018*). However, as the findings were qualitative in nature, they did not make possible use of AP characteristics to predict success or failure of ECC.

We combined imaging of ΔF/F with recordings of APs to gauge the success or failure of AP conduction and generation of a $Ca^{2+}$ transient over a distance of 1.6 mm (approximately threefold the length constant) along the length of mouse EDL muscle fibers (*Riisager et al., 2014*). APs peaking above –21 mV conducted and successfully triggered $Ca^{2+}$ release along the length of the fiber. The finding that APs peaking at –21 mV can still conduct was unexpected. However, the voltage dependence of Na channels is such that in retrospect this finding makes sense. Our previous loose patch voltage-clamp study in rat EDL revealed that by –20 mV ~ 90% of all non-inactivated $Na^+$ channels are activated (*Rich and Pinter, 2003*). Below –20 mV, there is a steep decline in the percentage of $Na^+$ channels activated. We hypothesize that once APs peak below –21 mV any decrement in the peak as they conduct along the fiber lessens activation of $Na^+$ channels and further reduces AP peak. This feedback loop leads to failure of propagation. Once APs peak above –21 mV, almost all available $Na^+$ channels are activated such that AP peak does not decline along the length of the fiber and conduction is successful. We conclude that AP peaks above –21 mV serve as a safety factor for ECC similar to the safety factor for synaptic transmission at the NMJ (*Rich, 2006*).

It has been suggested that failure of APs to propagate into t-tubules can contribute to reduction of $Ca^{2+}$ release from the SR (*Rassier and Minozzo, 2016*). We propose that conduction of APs along the length of fibers fails prior to failure of APs to conduct into t-tubules. This hypothesis is based on our finding that APs can be generated that trigger a local $Ca^{2+}$ signal in the region of the stimulating electrode. For this to occur, the AP must propagate into t-tubules near the stimulating electrode, but not along the length of the fiber. However, to definitively determine whether failure of APs to propagate into t-tubules is an early event in depolarization-induced failure of ECC, it would be necessary to directly measure t-tubule propagation.

### The integral of AP voltage predicts the local $Ca^{2+}$ transient

Our study and previous work suggest that $Ca^{2+}$ release from the SR begins to be triggered at voltages of –30 to –20 mV and becomes maximal at voltages above +10 mV (*Wang et al., 1999*; *Braubach et al., 2014*). Because APs peaking below –30 mV do not trigger elevation of $Ca^{2+}$, we set –30 mV as 0 and normalized AP peaks. Following normalization, the relationship between AP peak and the local $Ca^{2+}$ transient was good, but we wished to determine whether a closer relationship could be identified. It has previously been reported that depolarization of the resting potential triggers widening of APs (*Yensen et al., 2002*), and a recent computer simulation study suggested that widening of APs contributes to the increase in $Ca^{2+}$ signal that occurs following modest depolarization of the resting potential (*Senneff and Lowery, 2021*; see, however, *Yensen et al., 2002*). A previous study measured the area of the AP above –20 mV to estimate effects of changes in APs during repetitive stimulation on

contractility (*Riisager et al., 2014*). To incorporate consideration of both peak voltage and AP width, we took the integral of AP voltage with respect to time. This measure of APs closely correlated with the decrease in local $Ca^{2+}$ transient. AP integral did not correlate with the increase in $Ca^{2+}$ transient occurring with mild depolarization of the resting potential. One contributor to this discrepancy may be the elevation in resting $Ca^{2+}$ following depolarization of the resting potential (*Pedersen et al., 2019*).

## Distinct AP properties controlling conduction and $Ca^{2+}$ release

We propose that the AP parameter responsible for propagation is the peak voltage, whereas the AP property responsible for $Ca^{2+}$ release in each region of the fiber is the integral of AP peak with respect to time. This hypothesis is based on the physiology of the channels involved in each of these steps in ECC. Propagation of APs along the length of fibers is governed by $Na^+$ channels, which have rapid kinetics (*Mantegazza et al., 2021*). Because $Na^+$ channels activate rapidly, the duration of even narrow APs in one section of the fiber is more than sufficient to trigger opening of $Na^+$ channels in the adjacent region. Thus, widening of APs with depolarization of the resting potential would not be expected to enhance propagation. Our data is consistent with AP peak being the AP parameter responsible for successful conduction.

Release of $Ca^{2+}$ from the SR is governed by gating of Cav1.1 channels in the t-tubules (*Kovács et al., 1979*; *Rios and Brum, 1987*; *García et al., 1994*). Although movement of the Cav1.1 gating charges responsible for triggering opening of RyR1 and $Ca^{2+}$ release from the SR is faster than opening of Cav1.1 channels, the movement of the gating charges in $Ca_V1.1$ is relatively slow compared to the upstroke of the AP (*Schneider and Chandler, 1973*; *Banks et al., 2021*; *Savalli et al., 2021*; *Banks et al., 2022*) such that the wider the AP, the greater the charge movement until saturation is reached. Indeed, only ~65% of the total intramembrane charge is moved during an AP (*Banks et al., 2021*; *Banks et al., 2022*). Thus, the integral is the AP parameter responsible for local $Ca^{2+}$ release.

Despite being controlled by different aspects of the AP, depolarization-induced failure of conduction and failure to trigger $Ca^{2+}$ release occur almost simultaneously. While a localized $Ca^{2+}$ transient was often present at the time of failure of AP propagation, minimal additional depolarization of the resting potential caused failure of the local $Ca^{2+}$ transient as well. This is because widening of the AP can only compensate so much for reduction in AP peak. Once the AP peak is <–30 mV, no $Ca^{2+}$ release is triggered regardless of increases in AP width. Very little additional depolarization of the resting membrane potential is needed to reduce AP peaks from –20 mV to –30 mV.

## Inactivation of Nav1.4 rather than inactivation of SR $Ca^{2+}$ release appears to cause depolarization-induced failure of ECC

Recording in individual fibers during infusion of 16 mM $K^+$ resulted in higher AP peaks and thus larger $Ca^{2+}$ transients at a given resting potential than were obtained from sampling fibers from muscles perfused with different concentrations of $K^+$ (*Figure 1C* vs. *Figure 2E* and *Figure 1E* vs. *Figure 2F*). One potential explanation of this difference is the speed of depolarization. In recordings from individual fibers, infusion of 16 mM $K^+$ triggered an ~30 mV depolarization over several minutes. When sampling fibers from muscles in solutions with varying $K^+$ concentrations, muscles were incubated in each solution for 20 min prior to recording of APs. The prolonged depolarization allows for greater slow inactivation of $Na^+$ channels (*Ruff, 1996*; *Ruff, 1999*; *Rich and Pinter, 2003*), such that AP peak decreased at more negative resting potentials. In rat EDL muscle fibers, slow inactivation occurs at resting potentials more negative than fast inactivation (*Ruff, 1999*; *Rich and Pinter, 2003*). Thus, in situations where muscle fibers are chronically depolarized, slow inactivation plays a critical role in reducing excitability.

One factor that might contribute to failure of the $Ca^{2+}$ transient is inactivation of SR $Ca^{2+}$ release (*Ferreira Gregorio et al., 2017*; *Hernández-Ochoa and Schneider, 2018*). However, when 5 ms injections of current were used to trigger depolarization of inexcitable fibers with a mean resting potential of –52 mV, the average ΔF/F was ~90% as large as the maximal ΔF/F triggered by APs in the same fibers. While not definitive, these data suggest that inactivation of SR $Ca^{2+}$ release is a minor contributor to failure of ECC triggered by depolarization of the resting potential. We hypothesize that inactivation of Nav1.4 and the resultant reduction in AP peak is the primary factor responsible for depolarization-induced failure of ECC.

## Limitations

We used a voltage cutoff for the AP integral of –30 mV. This may not be the correct cutoff for all fibers. As shown in *Figure 2G*, there was a 15 mV range of AP peaks in different fibers at which there began to be a $Ca^{2+}$ transient. The use of the AP integral as described in this work is a simplification of the underlying processes. Both the magnitude and time course of gating charge movement in response to membrane potential changes are complex and nonlinear (*Ferreira Gregorio et al., 2017*). The $Ca^{2+}$ indicator used has slow kinetics and is high affinity. Saturation cannot be ruled out such that further work will need to be performed with dyes with more rapid kinetics and lower affinity to confirm conclusions regarding the relationship between AP properties and SR $Ca^{2+}$ release.

## Summary

Ours is the first study of excitation contraction coupling to establish quantitative relationships between resting potential, generation of APs, conduction of APs, and generation of $Ca^{2+}$ transients in individual fibers. Understanding these relationships provides the foundation for studies of failure of ECC triggered by depolarization of the resting potential.

## Methods

### Key resources table

| Reagent type (species) or resource | Designation | Source or reference | Identifiers | Additional information |
|---|---|---|---|---|
| Strain, strain background (*Mus musculus*) | GCAMP6f mice | Jackson Labs | Stock: cat #028865 crossed with cat# 030218 | |
| Chemical compound, drug | N-benzyl-*p*-toluenesulfonamide (BTS) | TCI America | Prod. #: B3082 | 0.05 mM |
| Software, algorithm | Spike2 | http://ced.co.uk/downloads/latestsoftware | | Version 8 |

### Mice

All animal procedures were performed in accordance with the policies of the Animal Care and Use Committee of Wright State University and were conducted in accordance with the United States Public Health Service's Policy on Humane Care and Use of Laboratory Animals.

Mice expressing GCAMP6f (*Chen et al., 2013*) in skeletal muscle were generated by crossing floxed GCAMP6f mice (Jackson Labs, B6J.Cg-*Gt(ROSA)26Sortm95.1(CAG-GCaMP6f)Hze*/MwarJ, cat #028865) with mice expressing parvalbumin promoter-driven Cre (Jackson Labs, B6.129P2-Pvalbtm1(cre)Arbr/J, cat# 030218). Mice were sacrificed using $CO_2$ or isoflurane inhalation followed by cervical dislocation.

### Solutions and temperature

All experiments were performed at 21–23°C within 4 hr of sacrifice. The same solutions were used for both force experiments and intracellular recording performed on EDL muscle fibers. The control solution contained (in mM) 118 NaCl, 3.5 KCl, 1.5 $CaCl_2$, 0.7 $MgSO_4$, 26.2 $NaHCO_3$, 1.7 $NaH2PO_4$, and 5.5 glucose and maintained at pH 7.3–7.4 by aeration with 95% $O_2$ and 5% $CO_2$. Solutions containing elevated concentrations of KCl (3.5, 10, 12, 14, and 16 mM) had corresponding reduction in NaCl (118, 111.5, 109.5, and 105.5 mM respectively).

### Ex vivo force measurements

The EDL muscle was dissected, and the proximal tendon was tied with a 6-0 caliber silk suture to a bar attached to a custom recording chamber. The distal tendon was tied to a hook and attached to the force transducer (Aurora Scientific). The EDL was stimulated with two platinum electrodes placed parallel to the muscle in the bath. The force transducer was controlled by a 305C two-channel controller (Aurora Scientific) and digitized by a Digidata 1550B digitizer (Molecular Devices). A S-900 pulse generator (Dagan) was used to generate 1 ms 100 V pulse. The pulse generator was triggered using pCLAMP 11 data acquisition and analysis software with a sampling frequency of 10 kHz. No

filtering was applied to the signal. Optimal length was determined by adjusting the tension of the muscle until maximal twitch force was achieved. During force recordings, the muscle was exposed to normal K$^+$ solution for 20 min, followed by high K$^+$ solution (10–16 mM) for 45 min, and then washed again with normal K$^+$ solution for 25 min to follow recovery. The EDL was stimulated with a twitch pulse every 5 min, and force was recorded.

### Ex vivo recordings of APs

To prevent contraction, muscles were loaded with 50 µM BTS (N-benzyl-p-toluenesulfonamide, Tokyo Chemical Industry, Tokyo, Japan, Cat#B3082) dissolved in DMSO for 45 min prior to recording. Muscle was stained with 10 µM 4-(4-diethylaminostyrl)-N-methylpyridinium iodide (Molecular Probes, discontinued). Muscle fibers were impaled with two sharp microelectrodes filled with 2 M potassium acetate solution containing 1 mM sulforhodamine 101 (Sigma-Aldrich, Cat#S7635) to allow for visualization. Resistances were between 15 and 30 MΩ, and capacitance compensation was optimized prior to recording. APs were evoked by a 0.2 ms injection of current ranging from 100 to 1000 nA. For recordings of APs during infusion of 16 mM K$^+$, after threshold was determined current injection was increased to 150% of threshold and perfusion of 16 mM K$^+$ was initiated. In studies of Ca$^{2+}$ release following failure of ECC, depolarization was achieved by increasing the duration of current injection to 5 ms. Fibers with resting potentials more depolarized than –74 mV in solution containing 3.5 mM KCl were discarded. Sampling frequency was 10 kHz with a 5 kHz-low pass filter.

### Imaging of ΔF/F

Muscle expressing GCAMP6f was imaged without staining (LeiCa$^{2+}$ I3 cube, band pass 450–490, long pass 515). Imaging was synchronized with triggering of APs using a Master-8 pulse generator (A.M.P.I., Jerusalem). Frames were acquired at 30 frames per second with a sCMOS camera (CS2100M-USB) using ThorCam software (Thorlab Inc, NJ). During infusion of solution containing high K$^+$, APs were triggered every 5 s. Each AP was synchronized with capture of 48 frames at 30 frames/s. Images were analyzed using ImageJ (NIH).

To confirm that the sampling rate of 30 Hz was not missing an early peak in Ca$^{2+}$ signal, we imaged a subset of fibers using a photomultiplier tube (Thorlab Inc) with a sampling rate of 5 kHz. The mean time to peak of the ΔF/F signal of APs in 3.5 mM Ca$^{2+}$ was 44.3 ± 3.2 ms (n = 9 muscles). This was slower than the time to peak of muscle twitch tension (22.3 ± 3.2 ms, n = 5 muscles), suggesting a slow rate of binding of Ca$^{2+}$ to GCAMP6f. The fluorescence signal 33 ms after triggering the AP averaged 96.2% ± 0.8% of the peak signal. These data suggest that the sampling frequency of 30 Hz introduced a slight underestimation of the peak ΔF/F.

For studies of the dependence of the Ca$^{2+}$ transient on AP properties, fibers were imaged using a ×40 objective and a single ROI 50–100 µm from the current injecting electrode was analyzed. For studies of conduction of APs along the length of the fiber, fibers were imaged using a ×5 objective. Placement of both electrodes in the same fiber was accomplished using the fluorescence signal triggered along the length of the fibers expressing GCAMP6f after triggering of an AP. ROIs were selected at regular lengths along the length of the muscle fiber being stimulated. The signal between stimuli in each ROI was used to record background, which was subtracted from peak signal detected after stimulation.

### Fitting of data with Boltzmann equations

Data for AP peak vs. resting potential, Ca$^{2+}$ image intensity vs. AP, and Ca$^{2+}$ image intensity vs. resting potential were all fit to a Boltzmann equation:

$$Out = LV + \frac{(HV-LV)}{1+e^{\frac{(V50-V)}{k}}} \qquad (1)$$

where *Out* represents the dependent variable (either AP peak or Ca$^{2+}$ image intensity), *V* is the independent voltage variable (either resting potential or AP peak), *LV* is the limiting value when *V* is very low (toward more negative), *HV* is the limiting value when *V* is very high (toward more positive), *V50* is the value of *V* at which *Out* is halfway between *HV* and *LV*, and *k* is the slope factor. All voltages and the variable *k* are expressed in mV, and Ca$^{2+}$ image intensity is in arbitrary units between 1 for maximum intensity for each experiment and 0.

## Statistics

Two types of statistical comparisons were made. For recordings at a single time point, repeated ANOVA with n as the number of mice was used. For comparisons within individual fibers followed over time during infusion of 16 mM $K^+$ and control fibers followed over time in 3.5 mM $K^+$, the paired Student's *t*-test was used with n as the number of fibers. The 12 fibers recorded from over time during infusion of 16 mM $K^+$ came from five different mice, and the 5 control fibers maintained in 3.5 mM $K^+$ came from three different mice. The numbers of animals and fibers for comparisons are described in the corresponding figure legends and text. All data are presented as mean ± SD. $p < 0.05$ was considered to be significant.

## Additional information

### Funding

| Funder | Grant reference number | Author |
|---|---|---|
| National Institute of Arthritis and Musculoskeletal and Skin Diseases | AR074985 | Mark M Rich |
| National Institute of Neurological Disorders and Stroke | R15NS099850 | Andrew A Voss |
| Muscular Dystrophy Association | 602459 | Mark M Rich |

The funders had no role in study design, data collection and interpretation, or the decision to submit the work for publication.

### Author contributions

Xueyong Wang, Formal analysis, Investigation, Methodology, Writing - original draft, Writing – review and editing; Murad Nawaz, Formal analysis, Investigation, Methodology, Writing - original draft; Chris DuPont, Jessica H Myers, Investigation, Writing – review and editing; Steve RA Burke, Investigation, Methodology, Writing – review and editing; Roger A Bannister, Conceptualization, Formal analysis, Writing – review and editing; Brent D Foy, Formal analysis, Software, Writing - original draft, Writing – review and editing; Andrew A Voss, Investigation, Supervision, Writing - original draft, Writing – review and editing; Mark M Rich, Conceptualization, Data curation, Formal analysis, Funding acquisition, Methodology, Project administration, Resources, Software, Supervision, Validation, Visualization, Writing - original draft, Writing – review and editing

### Author ORCIDs

Mark M Rich http://orcid.org/0000-0002-6956-5531

### Ethics

This study was performed in strict accordance with the recommendations in the Guide for the Care and Use of Laboratory Animals of the National Institutes of Health. All of the animals were handled according to guidelines from the IACUC committee of Wright State University. The protocol was approved by the IACUC Committee (Protocol #1179).

### Decision letter and Author response

Decision letter https://doi.org/10.7554/eLife.71588.sa1
Author response https://doi.org/10.7554/eLife.71588.sa2

## Additional files

### Supplementary files

• Transparent reporting form

## Data availability

All data generated during this study have been submitted to Dryad.

The following dataset was generated:

| Author(s) | Year | Dataset title | Dataset URL | Database and Identifier |
|-----------|------|---------------|-------------|-------------------------|
| Rich MM | 2021 | Effect of depolarization on action potentials and calcium transients in mouse skeletal muscle | https://doi.org/10.5061/dryad.sqv9s4n49 | Dryad Digital Repository, 10.5061/dryad.sqv9s4n49 |

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
