## [Editor Report]

This is a rigorous electrophysiological paper that investigates relationships between resting potential, action potential properties and conduction and Ca^2+^ transients. It makes an investigation of excitation contraction coupling failure associated with depolarization of the resting potential.

---

## [Decision Letter]

**Decision letter after peer review:**

Thank you for submitting your article "The Role of Action Potential Waveform in Failure of Excitation Contraction Coupling" for consideration by *eLife*. Your article has been reviewed by 3 peer reviewers, one of whom is a member of our Board of Reviewing Editors, and the evaluation has been overseen by Mone Zaidi as the Senior Editor. The reviewers have opted to remain anonymous.

Essential revisions:

All three reviewers responded positively to the study but had serious issues to raise concerning interpretation of the findings, the need for some further experiments, with a consequent large number of issues being raised as enumerated in the comments to follow. The reviewers have also indicated their reservations in broad terms as follows:

1) Reviewer 1 regards the observations as empirically helpful, but is very concerned about the further interpretation that need to address in particular (a) whether there is a possible tubular action potential conduction failure despite persistent surface membrane excitation and (b) the relative inactivation properties of Na channel function and of excitation contraction coupling.

2) To Reviewer 2 a large negative aspect of the manuscript in its current form was over-interpretation and the discussion about how these findings challenge fundamental aspect of our understanding of action potentials.

3) Although Reviewer 3 was enthusiastic about the study and the new finding (except for Figure 1) but made extensive and important comments. In particular they also felt that the conclusions are not fully justified. In particular there was a lack of information regarding the calcium indicator, commenting that due to the limitation of the calcium indicator that has been used, at least one other calcium indicator must be tested to confirm the conclusions.

4) Note that all three reviewers had independently raised such queries concerning the possibility of action potential failure, and interpretation of the all or none concept.

Despite the rather extensive changes that these points entail, the three reviewers felt they wished to give the authors the opportunity for further work and revisions, whilst clearly on the understanding that a revised version of the work would be subject to careful scrutiny in relationship to the recommendations that were made.

*Reviewer #1 (Recommendations for the authors):*

A) These are well conducted and elegant experiments, both in their electrical recordings and in their determinations of ca^2+^ transients. The findings provide a helpful and constructive empiric demonstration showing the point at which the processes of action potential activation and triggering of the ca^2+^ transient fail.

B) My problem is their taking this interpretation further, as their present experimental design does not consider:

1) The possibility of a persistent surface action potential generation despite a tubular action potential conduction failure. This will give the appearance of persistent regeneration Na channel activity reflecting the surface action potential wave, combined with an excitation contraction coupling failure. I wonder if changes of this kind are evidenced in the late component waveform changes as illustrated in Figure 1C.

2) If it is possible to resolve (1), the authors also need to draw attention to the presence of inactivation properties in both the Na channel mediating the action potential, and the excitation contraction coupling process, as evidence in inactivation properties of its underlying charge movement. Both these relationships are steep, and fall over similar voltage ranges.

3) Is it possible to obtain potentiometric optical records of tubular action potential propagation or its possible failure to demonstrate a persistence of a tubular action potential in the face of a failure in ca^2+^ transient activation?

4) Having addressed (3), one then needs to distinguish the inactivation properties and its voltage dependence respectively of Na channel inactivation [1] and the inactivation of excitation contraction coupling. The latter should be considered in terms of the corresponding inactivation properties of tension generation, the ca^2+^ transientand its underlying triggering intramembrane charge movement [2-3].

C) If it is possible to resolve the point (B) above, the outcome of this paper indeed distinguishes between action potential failure and excitation contraction coupling failure. If it is not possible to resolve this point, then it remains possible to report an empiric finding relating resting potential modification and skeletal muscle function.

References:

[1] Adrian RH, Marshall MW. Sodium currents in mammalian muscle (nih.gov)

[2] The effects of calcium deprivation upon mechanical and electrophysiological parameters in skeletal muscle fibres of the frog. – Lüttgau – 1979 – The Journal of Physiology – Wiley Online Library

[3] Analysis of 'off' tails of intramembrane charge movements in skeletal muscle of Rana temporaria. – Huang – 1984 – The Journal of Physiology – Wiley Online Library

*Reviewer #2 (Recommendations for the authors):*

The objective of this study is to further understand the mechanism by which increases in extracellular [K^+^] affect twitch force. On the one hand, Figure 1, which is about twitch force and action potential, brings nothing new because it has been shown many times. On the other hand, the data on the relationships between resting potential, action potential and ca^2+^ transient are new and very interesting. However, it must be expanded; and, at the present time there are serious major concerns that must be dealt with.

1. The ‘all or none’ concept. The authors are first basing this concept from a 1917 study, a concept that has been modified since then. They also claim that studies from the Renaud and Cairns groups described their K^+^ effects from an all or none perspective, which is not true. It is clear that K^+^-induced force depression involves graded decreases in both action potential peak, twitch and tetanic force, albeit the graded decreases are very steep but not in a all or none way.

In regard to action potential, the ‘all or none’ concept means that regardless of the stimulation strength, once threshold has been reached the action potential shape is always the same. However, this is true for a given physiological condition. When there is a change in the physiological condition, e.g., change in [Na+] or [K^+^], then the action potential shape will change according to how the kinetics of the Na^+^ channels are affected by the changes in ion concentration and/or membrane potential. However, in the new physiological condition, the ‘all or none’ concept remains.

So, the concept of all or none for the K^+^ effects must be entirely removed as it never existed and was never discussed in this way. Also remove the discussion about the definition of AP. The following definition is well accepted and data from this study and previous ones does not affect that definition: "An AP is a transient voltage change involving a depolarization phase followed by a repolarization phase. APs are all or none events within a given physiological definition, but can change shape when physiological conditions change.

2. Figure 1. The data in this figure has been shown in many past studies and absolutely provide no new information. Only Figure 1D is important to combine with Figure 2D and 3G. See more details in comments #7. All other data must be removed.

3. Another issue with Figure 1 and the entire studies is the experimental temperature of 25{degree sign}C when muscle temperatures during exercise exceed 37degC. Data at 25{degree sign}C are completely and physiologically irrelevant because the K^+^-induced force depression is highly temperature dependent, most studies showing a shift in the force-[K^+^] relationship shifted to higher [K^+^] as temperature is raised. Furthermore, the K^+^-induced twitch potentiation is much greater at 37{degree sign}C (Yensen study) than at 25{degree sign}C (Cairns study). In other words, no one can use the data in this study to understand the role of the K^+^ effects during fatigue. It is now possible to carry out experiments at higher temperature.

4. Ca^2+^ indicator. There are major differences between various ca^2+^ indicators providing µM to mM changes in intracellular [ca^2+^] during contraction, and the biggest problem is to estimate how ca^2+^ transient occurs during twitches. In this study GCAMP6f is used as a ca^2+^ indicator with its own limitations, but for which little is given and discussed, albeit references are provided. This is the most important and interesting part of the study and correlation between action potential and ca^2+^ transients have never been demonstrated. Here, the section need major strengthening as follows:

– The indicator is a protein being expressed in muscles. Where is the indicator mainly located? Indicators with nM affinity only measured [ca^2+^] in sarcomere, ca^2+^ values being in the µM range. However, indicators with very low affinity for ca^2+^ provide [ca^2+^] values close to mM. Why the differences? Most likely because [ca^2+^] near the releasing site must be very high to provide fast ca^2+^ diffusion into sarcomere and such concentration can only be measured with low ca^2+^ affinity indicator because those with nM affinity are saturated and tend to provide [ca^2+^] within the sarcomere. So to be relevant, one needs information about its location in the sarcoplasm.

– The authors mention on line 420 that the indicator has a high affinity but gives no values. To properly judge the possibility of saturation as raised by the authors one needs to know what is the affinity. For the results to be of any significance, that value must be provided as well as the conditions in which it was measured as test tube versus in fiber values are often different because of differences in solutions and myoplasm.

– It is obvious that GCAMP6f has a fast on-rate and a very slow off rate. These raises two questions. First, how long does it takes to reach the peak? Action potential overshoot occurs within 1 msec and time to twitch peak force of EDL is about 5-7 msec (37{degree sign}C). So, is the peak value of the ca^2+^ indicator occurring before or after the twitch time to peak?

– To strengthen the study, it would help if similar experiments were carried out with at least one other ca^2+^ indicator, perhaps one with very high affinity like Indo-1 and one with very low affinity to see what happens near the ca^2+^ releasing site as mentioned above.

All of these points should be discussed at the beginning of the discussion. The idea is not to dismiss the data. It is a first attempt and as long as the limitations are acceptable, the data are valuable. Also, in the Methods you should explain why pictures are taken every 33 msec (30 frames/sec). When one reads this, the reaction is that one cannot measured the ca^2+^ transient during a twitch in EDL that is shorter than 33 msec!

5. Why is a neighboring fiber is used for resting [ca^2+^]? Why not take a picture or two before fibers are stimulated to get the resting [ca^2+^]?

6. Fiber damage, lines 225-231. Why are the changes in resting EM and AP overshoot consider to be due to fiber damage? This question is raised because muscle movement during stimulation was blocked by BTS, so microelectrode damage is not expected. Another possibility is a leak around the microelectrodes. If this was the case, one would not get resting EM and overshoot as measured. The last thing is a continuous depolarization following microelectrode penetration. This is a major issue with KCl filled microelectrodes because K^+^ and Cl^-^ diffuses out of the microelectrodes while Cl^-^ leaves the fiber faster than K^+^ because of the huge Cl^-^ conductance of mammalian skeletal muscle fiber vs. that of K^+^. However, in this case the microelectrodes were filled with K^+^-acetate. Thus, the slow depolarization and overshoot decrease is more likely due to normal physiological changes associated with repetitive AP as it has been shown for example during a 2 sec 50 Hz tetanic stimulation of soleus muscle (Cairn study), which was stretched to reduce muscle movement during contraction. Regardless of the cause, what is important is to show what happens when [K^+^] is increased to 16 mM and demonstrate that the effects are much greater.

7. Slow inactivation. Another important aspect of the data presented in this study is the difference in AP overshoot-resting EM. In the discussion the authors raised the issue that for a given resting EM, AP overshoot is greater when measured at 16 mM K^+^ over a few minutes (Figure 2C) vs. after 45 min at elevated [K^+^] (Figure 1). I agree with the authors that the difference involved the slow inactivation process that takes several min to complete. Unfortunately, they raised the issue only in the discussion and are forcing us to discern the differences from different figures. I suggest that a figure is added to clearly show the overshoot-resting EM relationship in the same figure. This is clearly an important issue because we often look at a potential role of K^+^ in muscle fatigue during a 3-5 min fatigue period with the K^+^ effect over a 45 min period. Showing the differences in how depolarization and time affect overshoot is critical.

8. It is unfortunate that the authors did not think about measuring [ca^2+^] when they measured AP after inducing membrane depolarizations at high [K^+^]. There is evidence for slow inactivation of the ca^2+^ release mechanism involving both CaV1.1 and RyR1. So, it would be interesting to see if prolonged K^+^-induced membrane depolarization also affects over time the ca^2+^ transient; i.e., different [ca^2+^]-resting EM or AP relationships. This would really increase the interest for this study.

9. Statistics. The greater the number of paired t-test the lower the overall significance; i.e., each test may be with P < 0.05, but too many t-tests and the overall probability becomes greater than 0.05. ANOVA are much stronger and must be used. An n=3 such as in line 174 is a poor sample size for statistics.

10. How were resting EM and overshoot values averaged? One cannot take an average from all tested fibers from different muscles. One must average the values from fibers from the same muscles and then calculated the final mean from the muscle averages. Sample size should then be number of muscles/number of fibers. Statistics should also use the muscle averages unless a repeated ANOVA is used.

11. Lines 175-177: this is an example of discussion in the Results section. Remove all of them and keep them for the Discussion section.

12. Figure 1C: Stimulation artifacts are quite large and AP starts before EM is back to pre-stimulation. Consequently, overshoots are overestimated. Greater distances are necessary between the injecting current microelectrode and EM recording microelectrode. Also, if current was injected to trigger APs, why is the artifacts so large?

13. It would also be interesting to compare the AP overshoot-resting EM with steady state fast and slow inactivation curve of NaV1.4 channels to further emphasize the similarity and the physiological relevance of these inactivation curves.

14. Pedersen and others often explain the increase in twitch [ca^2+^] with repetitive stimulation by an increase in [ca^2+^] between contractions because it means that more ca^2+^ remains bound to ca^2+^ binding proteins so when ca^2+^ is released more remains in the fee state. You do have an increase in resting [ca^2+^] in Figure 3C and this possibility must be discussed in the Discussion because it can be a reason as to why there is a portion of the ca^2+^ transient – overshoot that cannot be explained by the overshoot decrease.

15. Lines 394. A non-linear relationship does not imply a poor correlation. You have in fact obtained excellent relationships, they are simply not linear. For such relationships you must fit the data for nonlinear relations and you will still get excellent correlation coefficient. What you are looking for is a linear relationship. Correct the discussion.

*Reviewer #3 (Recommendations for the authors):*

This manuscript is a very interesting piece of work, and it leads to a very important conclusion regarding SR ca^2+^ release. It shows that the release is not as binary as it is often portrayed in textbooks and by colleagues in our field. Rather this work demonstrates that the SR ca^2+^ release is highly dependent on the waveform of the activating action potential. This finding may be relevant for fatigue and for muscle weakness in some paralytic disorders.

The experimental approach is interesting, and experiments are well performed. Moreover, the authors are aware of some of the limitations and these limitations are mentioned. They are, however, not really taken into consideration when the data is interpretated. In the mind of the reviewer this leads to some degree of over-interpretation of the data. Two particular elements of concern exist that should be modified/addressed:

First, the discussion of whether the signals are all-or-none is misguided by the fact that the two electrodes are close to each other. While not mentioned in the methods (please correct this) the electrodes appear to be inserted at very close proximity and the ca^2+^ signals are (presumably) recorded within the same region where the electrodes are impaled. This means that it is not possible to exclude that the compromised action potentials at high K^+^ did not propagate along the fiber. It is therefore hard to exclude that some of the loss of force at elevated K^+^ is caused by all-or-none behavior due to failing propagation. It is recommended that statements like in L195-199 are modified because how do you know that the signals propagated? If they did not propagate the all-or-none is more accurate description perhaps. Please also re-consider that the signals recorded at the excitation zone naturally will depend on the size of the passive response to the action potential trigger current and hence the recorded signals with the voltage electrode will depend on both the regenerative action potential signals and the triggering current. In propagating signals, only the true action potential will remain and the passive response to the action potential triggering current would disappear.

The paper is interesting due to the data and in the mind of the reviewer the whole narrative about these findings leading to new definition of an action potential and the challenge of the all-or-none theory is not really needed. Frankly, it takes away the focus of the nice scientific data that demonstrates that SR ca^2+^ release is strongly dependent of the waveform of the action potential. This is new and sufficiently novel to justify the work.

Second, the imaging of ca^2+^ is with a high affinity reporting system and with low sampling frequency. These two experimental design characteristics means that the peak of the ca^2+^ transients could have been missed, and it is therefore difficult to know how the transient peaks were affected when the action potential peaks declined. The conclusion that the waveform integral above -30 mV is the best predictor for ca^2+^ release is likely correct, but again the conclusions are rather dogmatic and should be more balanced taking into consideration the limitations of the method.

---

## [Author Response]

Essential revisions:Reviewer #1 (Recommendations for the authors):A) These are well conducted and elegant experiments, both in their electrical recordings and in their determinations of ca^2+^ transients. The findings provide a helpful and constructive empiric demonstration showing the point at which the processes of action potential activation and triggering of the ca^2+^ transient fail.B) My problem is their taking this interpretation further, as their present experimental design does not consider:1) The possibility of a persistent surface action potential generation despite a tubular action potential conduction failure. This will give the appearance of persistent regeneration Na channel activity reflecting the surface action potential wave, combined with an excitation contraction coupling failure. I wonder if changes of this kind are evidenced in the late component waveform changes as illustrated in Figure 1C.

This is a very insightful comment and caused us to perform additional experiments to determine whether failure of AP conduction along the length of fibers might contribute to failure of ECC. We have added 2 new figures showing these data. Our new data suggest failure of conduction along the length of the fiber rather than failure of AP conduction into t-tubules is the cause of failure of ECC.

2) If it is possible to resolve (1), the authors also need to draw attention to the presence of inactivation properties in both the Na channel mediating the action potential, and the excitation contraction coupling process, as evidence in inactivation properties of its underlying charge movement. Both these relationships are steep, and fall over similar voltage ranges.

We performed additional experiments to determine whether inactivation of Na channels or inactivation of SR Ca release is the limiting factor in depolarization-induced failure of ECC. Our data suggests inactivation of Na channels and failure of APs to trigger ca^2+^ release occurs prior to inactivation of SR ca^2+^ release. These data are presented in a new figure.

3) Is it possible to obtain potentiometric optical records of tubular action potential propagation or its possible failure to demonstrate a persistence of a tubular action potential in the face of a failure in ca^2+^ transient activation?

We believe the new data presented make this possibility less likely, so we opted not to perform experiments with potentiometric dyes.

4) Having addressed (3), one then needs to distinguish the inactivation properties and its voltage dependence respectively of Na channel inactivation [1] and the inactivation of excitation contraction coupling. The latter should be considered in terms of the corresponding inactivation properties of tension generation, the ca^2+^ transientand its underlying triggering intramembrane charge movement [2-3].

New experiments have been performed and a new figure has been generated. Inactivation of Na channels appears to be responsible for failure of ECC.

C) If it is possible to resolve the point (B) above, the outcome of this paper indeed distinguishes between action potential failure and excitation contraction coupling failure. If it is not possible to resolve this point, then it remains possible to report an empiric finding relating resting potential modification and skeletal muscle function.

We believe the new experiments performed have greatly strengthened our study and support our conclusion that depolarization-induced failure of ECC is due to failure of AP generation/conduction rather than inactivation of SR Ca release.

Reviewer #2 (Recommendations for the authors):The objective of this study is to further understand the mechanism by which increases in extracellular [K^+^] affect twitch force. On the one hand, Figure 1, which is about twitch force and action potential, brings nothing new because it has been shown many times. On the other hand, the data on the relationships between resting potential, action potential and ca^2+^ transient are new and very interesting. However, it must be expanded; and, at the present time there are serious major concerns that must be dealt with.1. The ‘all or none’ concept. The authors are first basing this concept from a 1917 study, a concept that has been modified since then. They also claim that studies from the Renaud and Cairns groups described their K^+^ effects from an all or none perspective, which is not true. It is clear that K^+^-induced force depression involves graded decreases in both action potential peak, twitch and tetanic force, albeit the graded decreases are very steep but not in a all or none way.In regard to action potential, the ‘all or none’ concept means that regardless of the stimulation strength, once threshold has been reached the action potential shape is always the same. However, this is true for a given physiological condition. When there is a change in the physiological condition, e.g., change in [Na+] or [K^+^], then the action potential shape will change according to how the kinetics of the Na^+^ channels are affected by the changes in ion concentration and/or membrane potential. However, in the new physiological condition, the ‘all or none’ concept remains.So, the concept of all or none for the K^+^ effects must be entirely removed as it never existed and was never discussed in this way. Also remove the discussion about the definition of AP. The following definition is well accepted and data from this study and previous ones does not affect that definition: "An AP is a transient voltage change involving a depolarization phase followed by a repolarization phase. APs are all or none events within a given physiological definition, but can change shape when physiological conditions change.

The all or none concept has been removed and the discussion of what is an AP has been removed.

2. Figure 1. The data in this figure has been shown in many past studies and absolutely provide no new information. Only Figure 1D is important to combine with Figure 2D and 3G. See more details in comments #7. All other data must be removed.

We agree with the reviewer that the force data confirm previous results. We left the animal numbers for force experiments at 3 as they are confirmatory. We have redone all the intracellular recordings with higher n and with simultaneous Ca imaging. We hope the reviewer will agree that the new Figure 1 substantially moves the field forward.

3. Another issue with Figure 1 and the entire studies is the experimental temperature of 25{degree sign}C when muscle temperatures during exercise exceed 37degC. Data at 25{degree sign}C are completely and physiologically irrelevant because the K^+^-induced force depression is highly temperature dependent, most studies showing a shift in the force-[K^+^] relationship shifted to higher [K^+^] as temperature is raised. Furthermore, the K^+^-induced twitch potentiation is much greater at 37{degree sign}C (Yensen study) than at 25{degree sign}C (Cairns study). In other words, no one can use the data in this study to understand the role of the K^+^ effects during fatigue. It is now possible to carry out experiments at higher temperature.

We agree and plan to perform these experiments in the future. We are working to modify our set up, and hope within the next year to begin experiments closer to physiologic temperature.

4. Ca^2+^ indicator. There are major differences between various ca^2+^ indicators providing µM to mM changes in intracellular [ca^2+^] during contraction, and the biggest problem is to estimate how ca^2+^ transient occurs during twitches. In this study GCAMP6f is used as a ca^2+^ indicator with its own limitations, but for which little is given and discussed, albeit references are provided. This is the most important and interesting part of the study and correlation between action potential and ca^2+^ transients have never been demonstrated. Here, the section need major strengthening as follows:– The indicator is a protein being expressed in muscles. Where is the indicator mainly located? Indicators with nM affinity only measured [ca^2+^] in sarcomere, ca^2+^ values being in the µM range. However, indicators with very low affinity for ca^2+^ provide [ca^2+^] values close to mM. Why the differences? Most likely because [ca^2+^] near the releasing site must be very high to provide fast ca^2+^ diffusion into sarcomere and such concentration can only be measured with low ca^2+^ affinity indicator because those with nM affinity are saturated and tend to provide [ca^2+^] within the sarcomere. So to be relevant, one needs information about its location in the sarcoplasm.

We are not attempting to determine the ca^2+^ concentration in muscle fibers. We cannot rule out that GCAMP6F might be saturating. We have added a limitations section to the discussion where we discuss limitations of the ca^2+^ dye.

– The authors mention on line 420 that the indicator has a high affinity but gives no values. To properly judge the possibility of saturation as raised by the authors one needs to know what is the affinity. For the results to be of any significance, that value must be provided as well as the conditions in which it was measured as test tube versus in fiber values are often different because of differences in solutions and myoplasm.

We have added the value. It was derived in situ in cardiomyocytes.

– It is obvious that GCAMP6f has a fast on-rate and a very slow off rate. These raises two questions. First, how long does it takes to reach the peak? Action potential overshoot occurs within 1 msec and time to twitch peak force of EDL is about 5-7 msec (37{degree sign}C). So, is the peak value of the ca^2+^ indicator occurring before or after the twitch time to peak?

We now include data on time to peak of the GCAMP6f signal and found it is near 40 ms. We measured the time to peak of force measurements at 22C and found that for the EDL it was close to 22 ms. These data suggest that GCAMP6f has a slow on rate as well as having a slow off rate and we conclude the peak of the fluorescence signal is occurring after the twitch time to peak. These data are now included in the methods section.

– To strengthen the study, it would help if similar experiments were carried out with at least one other ca^2+^ indicator, perhaps one with very high affinity like Indo-1 and one with very low affinity to see what happens near the ca^2+^ releasing site as mentioned above.

We plan to do this in a follow up study, but given the extensive additional experiments performed, we believe this is beyond the scope of the current manuscript.

All of these points should be discussed at the beginning of the discussion. The idea is not to dismiss the data. It is a first attempt and as long as the limitations are acceptable, the data are valuable. Also, in the Methods you should explain why pictures are taken every 33 msec (30 frames/sec). When one reads this, the reaction is that one cannot measured the ca^2+^ transient during a twitch in EDL that is shorter than 33 msec!

We performed some experiments using a PMT tube at a sampling rate of 5 kHZ. We now include data on time to peak and the small error introduced by our 30 Hz sample rate with the camera in the methods section. The reason for the 30 Hz sampling rate is that this is the fastest rate at which we can take pictures with our camera.

5. Why is a neighboring fiber is used for resting [ca^2+^]? Why not take a picture or two before fibers are stimulated to get the resting [ca^2+^]?

We have corrected this and now use the same fiber for background. This did not change results.

6. Fiber damage, lines 225-231. Why are the changes in resting EM and AP overshoot consider to be due to fiber damage? This question is raised because muscle movement during stimulation was blocked by BTS, so microelectrode damage is not expected. Another possibility is a leak around the microelectrodes. If this was the case, one would not get resting EM and overshoot as measured. The last thing is a continuous depolarization following microelectrode penetration. This is a major issue with KCl filled microelectrodes because K^+^ and Cl^-^ diffuses out of the microelectrodes while Cl^-^ leaves the fiber faster than K^+^ because of the huge Cl^-^ conductance of mammalian skeletal muscle fiber vs. that of K^+^. However, in this case the microelectrodes were filled with K^+^-acetate. Thus, the slow depolarization and overshoot decrease is more likely due to normal physiological changes associated with repetitive AP as it has been shown for example during a 2 sec 50 Hz tetanic stimulation of soleus muscle (Cairn study), which was stretched to reduce muscle movement during contraction. Regardless of the cause, what is important is to show what happens when [K^+^] is increased to 16 mM and demonstrate that the effects are much greater.

We agree with the reviewer that it is not critical for our study whether depolarization is induced by damage or high K^+^. We have shortened discussion of damage-induced depolarization and have removed it from the figure.

7. Slow inactivation. Another important aspect of the data presented in this study is the difference in AP overshoot-resting EM. In the discussion the authors raised the issue that for a given resting EM, AP overshoot is greater when measured at 16 mM K^+^ over a few minutes (Figure 2C) vs. after 45 min at elevated [K^+^] (Figure 1). I agree with the authors that the difference involved the slow inactivation process that takes several min to complete. Unfortunately, they raised the issue only in the discussion and are forcing us to discern the differences from different figures. I suggest that a figure is added to clearly show the overshoot-resting EM relationship in the same figure. This is clearly an important issue because we often look at a potential role of K^+^ in muscle fatigue during a 3-5 min fatigue period with the K^+^ effect over a 45 min period. Showing the differences in how depolarization and time affect overshoot is critical.

We agree with the reviewer that this is a critical issue. However, we have opted not to put the data in the same figure as we could not figure out how to do so without disrupting the flow of the manuscript. We hope that pointing the difference in the discussion will suffice.

8. It is unfortunate that the authors did not think about measuring [ca^2+^] when they measured AP after inducing membrane depolarizations at high [K^+^]. There is evidence for slow inactivation of the ca^2+^ release mechanism involving both CaV1.1 and RyR1. So, it would be interesting to see if prolonged K^+^-induced membrane depolarization also affects over time the ca^2+^ transient; i.e., different [ca^2+^]-resting EM or AP relationships. This would really increase the interest for this study.

We re-did all of the 20-40 minutes depolarization experiments in muscle expressing GCAMP6f. We present these new data in a new version of Figure 1 that includes ca^2+^ imaging.

9. Statistics. The greater the number of paired t-test the lower the overall significance; i.e., each test may be with P < 0.05, but too many t-tests and the overall probability becomes greater than 0.05. ANOVA are much stronger and must be used. An n=3 such as in line 174 is a poor sample size for statistics.

We use n of 3 mice for the force experiments as they are confirmatory of previous excellent studies and are not meant to stand alone. Rather than eliminate these confirmatory data we have made their presentation as brief as possible. For most other statistics we have at least 5 mice as the n as used nested analysis of variance. In 14 and 16 mM K^+^ where there is no evidence of muscle contraction n was less than 5 mice.

10. How were resting EM and overshoot values averaged? One cannot take an average from all tested fibers from different muscles. One must average the values from fibers from the same muscles and then calculated the final mean from the muscle averages. Sample size should then be number of muscles/number of fibers. Statistics should also use the muscle averages unless a repeated ANOVA is used.

We use the number of muscles as sample size for comparisons between muscles and average the values from muscles to get the final average. We used a repeated ANOVA as described in the methods. We have now clarified this. We have lengthened the statistics section to more clearly state how the different types of data were analyzed.

11. Lines 175-177: this is an example of discussion in the Results section. Remove all of them and keep them for the Discussion section.

We have tried to keep discussion topics in the discussion.

12. Figure 1C: Stimulation artifacts are quite large and AP starts before EM is back to pre-stimulation. Consequently, overshoots are overestimated. Greater distances are necessary between the injecting current microelectrode and EM recording microelectrode. Also, if current was injected to trigger APs, why is the artifacts so large?

We have added new experiments with the electrodes far apart. These data are part of 2 new figures.

13. It would also be interesting to compare the AP overshoot-resting EM with steady state fast and slow inactivation curve of NaV1.4 channels to further emphasize the similarity and the physiological relevance of these inactivation curves.

Given the large number of new experiments performed we did not also measure the voltage dependence of fast and slow inactivation of Na^+^ channels in the mouse EDL for this study. We have previously measured the voltage dependence of fast and slow inactivation in the rat EDL muscle using the loose patch clamp technique. The reference for these studies is now included.

14. Pedersen and others often explain the increase in twitch [ca^2+^] with repetitive stimulation by an increase in [ca^2+^] between contractions because it means that more ca^2+^ remains bound to ca^2+^ binding proteins so when ca^2+^ is released more remains in the fee state. You do have an increase in resting [ca^2+^] in Figure 3C and this possibility must be discussed in the Discussion because it can be a reason as to why there is a portion of the ca^2+^ transient – overshoot that cannot be explained by the overshoot decrease.

We now include mention of a recent modeling study, which suggests the increase in ca^2+^ signal is due to widening of the AP. We also mention the possibility that an increase in resting ca^2+^ is the cause.

15. Lines 394. A non-linear relationship does not imply a poor correlation. You have in fact obtained excellent relationships, they are simply not linear. For such relationships you must fit the data for nonlinear relations and you will still get excellent correlation coefficient.What you are looking for is a linear relationship. Correct the discussion.

Done.

Reviewer #3 (Recommendations for the authors):This manuscript is a very interesting piece of work, and it leads to a very important conclusion regarding SR ca^2+^ release. It shows that the release is not as binary as it is often portrayed in textbooks and by colleagues in our field. Rather this work demonstrates that the SR ca^2+^ release is highly dependent on the waveform of the activating action potential. This finding may be relevant for fatigue and for muscle weakness in some paralytic disorders.The experimental approach is interesting, and experiments are well performed. Moreover, the authors are aware of some of the limitations and these limitations are mentioned. They are, however, not really taken into consideration when the data is interpretated. In the mind of the reviewer this leads to some degree of over-interpretation of the data. Two particular elements of concern exist that should be modified/addressed:First, the discussion of whether the signals are all-or-none is misguided by the fact that the two electrodes are close to each other. While not mentioned in the methods (please correct this) the electrodes appear to be inserted at very close proximity and the ca^2+^ signals are (presumably) recorded within the same region where the electrodes are impaled. This means that it is not possible to exclude that the compromised action potentials at high K^+^ did not propagate along the fiber. It is therefore hard to exclude that some of the loss of force at elevated K^+^ is caused by all-or-none behavior due to failing propagation. It is recommended that statements like in L195-199 are modified because how do you know that the signals propagated? If they did not propagate the all-or-none is more accurate description perhaps. Please also re-consider that the signals recorded at the excitation zone naturally will depend on the size of the passive response to the action potential trigger current and hence the recorded signals with the voltage electrode will depend on both the regenerative action potential signals and the triggering current. In propagating signals, only the true action potential will remain and the passive response to the action potential triggering current would disappear.The paper is interesting due to the data and in the mind of the reviewer the whole narrative about these findings leading to new definition of an action potential and the challenge of the all-or-none theory is not really needed. Frankly, it takes away the focus of the nice scientific data that demonstrates that SR ca^2+^ release is strongly dependent of the waveform of the action potential. This is new and sufficiently novel to justify the work.

These are excellent comments that we took to heart. We have performed new experiments examining conduction along the length of the fiber. This has led to the addition of 2 new figures. We have also removed discussion of all-or-none APs.

Second, the imaging of ca^2+^ is with a high affinity reporting system and with low sampling frequency. These two experimental design characteristics means that the peak of the ca^2+^ transients could have been missed, and it is therefore difficult to know how the transient peaks were affected when the action potential peaks declined. The conclusion that the waveform integral above -30 mV is the best predictor for ca^2+^ release is likely correct, but again the conclusions are rather dogmatic and should be more balanced taking into consideration the limitations of the method.

We have performed new experiments using a PMT tube, which provides for a 5 kHz sampling rate. These experiments confirmed that errors introduced by our slow sampling rate with the camera are modest. We have added a section to the discussion where we mention some of the limitations of our ca^2+^ imaging technique.